# Through-Holes Design for Ideal LiNbO_3_ A1 Resonators

**DOI:** 10.3390/mi14071341

**Published:** 2023-06-30

**Authors:** Shu-Mao Wu, Chen-Bei Hao, Zhen-Hui Qin, Yong Wang, Hua-Yang Chen, Si-Yuan Yu, Yan-Feng Chen

**Affiliations:** 1National Laboratory of Solid State Microstructures, Department of Materials Science and Engineering, Nanjing University, Nanjing 210093, China; shumaowu@smail.nju.edu.cn (S.-M.W.); haochenbei@smail.nju.edu.cn (C.-B.H.); qinzhenhui@smail.nju.edu.cn (Z.-H.Q.); mf21340034@smail.nju.edu.cn (H.-Y.C.); 2State Key Lab of Crystal Materials, Shandong University, Jinan 250100, China; yongwang@email.cn

**Keywords:** wireless communications, LiNbO_3_ thin films, XBARs, Lamb wave resonators/filters

## Abstract

This paper proposes a method to realize ideal lithium niobate (LiNbO_3_) A1 resonators. By introducing subwavelength through-holes between the interdigital transducer (IDT) electrodes on the LiNbO_3_ surface, all unfavorable spurious modes of the resonators can be suppressed completely. It is convenient and valid for various IDT electrode parameters and different LiNbO_3_ thicknesses. Also, this method does not require additional device fabrication steps. At the same time, these through-holes can greatly reduce the suspended area of the LiNbO_3_ thin film, thus significantly improving the design flexibility, compactness, mechanical stability, temperature stability, and power tolerance of the resonators (and subsequent filters). It is expected to become an important means to promote the practical application of LiNbO_3_ A1 filters and even all Lamb waves filters.

## 1. Introduction

Acoustic filters in microwave frequencies have a long and extensive application in mobile communications [1,2,3]. In the third and fourth-generation mobile communications (3G and 4G), operating frequencies of the relevant acoustic filters have reached over 3 GHz. With the rapid development of the new generation of communication technology (5G/6G/WiFi6/WiFi7), the demand for more compact, higher frequency, and larger bandwidth filters is increasing rapidly, such as various new acoustic filters working in n77, n78, n79 WiFi6E, and Wifi7 bands [4,5,6].

The existing mainstream commercial microwave acoustic filters mainly adopt two technical paths [7]. One is surface acoustic wave (SAW) filters based on LiNbO_3_ and LiTaO_3_ single crystals. Such devices have been developed, e.g., TC-SAW [8,9], I.H.P-SAW [10], TF-SAW [11], and Super-SAW [12,13]. Their frequencies have reached over 3.7 GHz and can realize large bandwidths to, e.g., 730 MHz [14]. However, the operating frequencies of such devices are limited by the lithographic line width to prepare interdigital transducers (IDTs), so further development to higher frequency will lead to high cost and technical difficulty, Ref. [15]. The other path is bulk acoustic wave (BAW) filters based on aluminum nitride (AlN) piezoelectric films. Their advantage is that their frequencies are not affected by the lithographic line width, and it is relatively easy to realize a filter over 3.5 GHz [16,17,18,19]. However, the bandwidths of the traditional BAW filters are only around 3% due to the low electromechanical coupling coefficient (about ~6%) of the AlN films [20]. Recently, the bandwidth of some novel BAW filters has been significantly improved by doping scandium (Sc) to increase the electromechanical coupling coefficient of AlN [21,22,23]. For example, Qorvo recently announced a high-performance BAW filter operating at 8 GHz with a bandwidth greater than 6% [24].

In addition to longitudinal BAWs, acoustic modes in films also include Lamb waves and shear horizontal (SH) waves. Lamb waves further include symmetric (S) modes and antisymmetric (A) modes. Among them, thickness-shear modes (e.g., A1, S2, A3, S4 …) are particularly beneficial for realizing high-frequency, large-bandwidth acoustic filters. Firstly, the excitation of such modes of several GHz or even higher frequencies can be achieved with only micron-scale linewidth IDTs. Secondly, using LiNbO_3_ thin films, such modes (especially A1) can have high electromechanical coupling coefficients, facilitating the implementation of ultra-large bandwidth filters.

Around 2010, researchers from Murata first prepared A1 resonators based on LiNbO_3_ thin films [25]. Their A1 resonators are at 4.5/6.3 GHz using CVD-grown LiNbO_3_ films with a thickness of 500 nm. However, the single crystallinity of LiNbO_3_ obtained by the CVD method is poor, leading to lower quality factors (Q value) of the resonators. With the advancement of high-quality LiNbO_3_ single crystal thin films, A1 resonators have entered a stage of rapid development [26]. In 2017, Yang et al. utilized Z-cut LiNbO_3_ thin films to achieve A1 resonators with operating frequencies covering 1–5 GHz and electromechanical coupling coefficients exceeding 26% [27,28,29] and by using them to achieve filters with bandwidths up to 10%. Lu et al. used 128° Y-cut LiNbO_3_ thin films to further increase the electromechanical coupling coefficient of the A1 resonators to 46% in similar frequencies [30,31], which are expected to realize filters with bandwidth over 20%. During the same time, Plessky et al. implemented similar A1 resonators on Z-cut LiNbO_3_ film and named it XBAR [32,33]. Based on the XBAR technology, Resonant has implemented a series of filters with extremely large bandwidths operating over 3.5 GHz [34]. So only one filter is needed to meet the filtering requirements of WiFi-6E (i.e., large bandwidth filtering from 6.0 GHz to 7.2 GHz) [35].

Another advantage of A1 resonators is that their operating frequencies primarily depend on the thickness of the piezoelectric films [36]. By local thinning of the LiNbO_3_ thin films, high-frequency and large-bandwidth A1 filters can be conveniently designed [37,38]. With the reduction of LiNbO_3_ thickness to below 200 nm, A1 filters are expected to operate at frequencies above 10 GHz while maintaining the relative bandwidth [39].

Acoustic filters in microwave frequencies are generally implemented by multiple resonators connected in series and parallel. As mentioned above, LiNbO_3_ A1 resonators enable high-frequency and ultra-large bandwidth A1 filters. However, in almost all A1 resonator designs reported so far, a higher-order spurious mode readily appears at a frequency slightly above the anti-resonant peak. This spurious mode necessarily causes significant fluctuations in the A1 filter passband, degrading the filtering performance.

To suppress or avoid this annoying spurious mode, Plessky et al. proposed a scheme to adjust the frequency difference between the main mode of the resonator and this spurious mode by changing the electrode pitch. In certain bandwidth-specific scenarios, this approach can move the spurious mode outside the filters’ passbands [40,41]. Koulakis et al. found that this spurious mode in the XBARs can be partially suppressed by setting IDT electrodes to an appropriate thickness [35]. Yang et al. found that high-order spurious modes could be better suppressed by increasing the electrode pitches and picking suitable numbers of electrodes. Their further research showed that dispersion matching is the key factor for suppressing high-order spurious modes [42]. They etched grooves with a certain depth on the LiNbO_3_ surface and deposited metal electrodes in the grooves, effectively suppressing the spurious modes [43].

To sum up, whether it is to suppress or avoid the unwanted spurious mode, the existing solutions either require additional device preparation steps or have strict restrictions on device design. In this work, as a contribution to A1 resonators, we will demonstrate a general and simple design method to completely suppress this spurious mode, i.e., introducing subwavelength through-holes among the IDT electrodes on the LiNbO_3_ surface. Using a Z-cut LiNbO_3_ A1 resonator as an example, we demonstrate through three-dimensional (3D) multi-physics simulations that all A1 resonators with the through-holes have a spurious mode-free conductance response for various design parameters (including holes spacing/position, IDT apertures, electrode duty ratio, and LiNbO_3_ thicknesses). More advantageously, for A1 resonators (even all Lamb wave resonators) fabricated by a surface etching process, the through-holes can greatly reduce the suspended area of piezoelectric thin film, thus facilitating design convenience, mechanical stability, temperature stability, and compactness of the resonators/filters.

## 2. A1 Resonators Based on LiNbO_3_ Thin Film

A typical LiNbO_3_ A1 resonator is shown in Figure 1. IDT placed on the surface of the suspended LiNbO_3_ thin film is used to excite the A1 mode.

In order to obtain a suspended structure, it is generally necessary to etch through the LiNbO_3_ thin film to form release windows at the regions between the IDT electrodes and the Bus electrodes. Afterward, silicon (Si) or silicon oxide (SiO_2_) at the bottom of the LiNbO_3_ thin film is removed by wet or dry etching.

In the A1 resonators, acoustic energy is mainly concentrated in the transverse center of the resonators, with relatively minimal energy in the regions on either side. Typically, this phenomenon becomes more pronounced as the number of IDT electrodes increases [31,42]. Our study also verified this phenomenon. Figure 1c shows the distribution of the displacement field in the out-of-plane direction at the A1 resonating frequency. It can be seen that almost all vibrations of the A1 mode are in the region between the IDT electrodes, with virtually no vibration in the regions on either side. This is another advantage of the A1 resonators (also applicable to other thickness modes, e.g., A3). During the fabrication of the resonators, unlike non-thickness modes (e.g., S0 and SH0) resonators [44,45,46], the A1 resonators are less demanding on the broadening, sidewall inclination and roughness of the suspended LiNbO_3_ thin film boundary, offering considerable process convenience.

As mentioned earlier, the higher-order spurious modes in A1 resonators have a crucial relationship with the electrode design. In the previous designs, it is especially closely related to the electrode duty ratio. The duty ratio depends on the electrode width (We) and the electrode pitch (G), and a small ratio (generally < 0.15) is needed for the suppression of spurious modes. In previous approaches, reducing We and/or increasing G are mainly used to obtain a small duty ratio. However, overly large G (e.g., over 10 μm) will increase the device area, especially for resonators with a large number of electrodes. Also, the substrate/box-layer etching (release) time increases significantly, making the resonators more difficult to process. In addition, an excessively large suspension area of the LiNbO_3_ thin film will impair the stability and miniaturization of the resonators. On the other hand, with small We (e.g., less than 1 μm), although the spurious modes can be suppressed by moderate G, the requirements for lithographic precision are significantly increased. Therefore, it is challenging to achieve spurious mode-free A1 resonators with the smaller G and larger We.

Aiming at this problem, we propose a method to realize ideal LiNbO_3_ A1 resonators. By introducing subwavelength through-holes between IDT electrodes with large widths and small pitches, the spurious modes of resonators can be suppressed perfectly. As an example, our A1 resonators are prepared on a 300 nm LiNbO_3_ thin film, where the IDTs are modes of Gold (Au) with a thickness of 50 nm. Key parameters of our A1 resonators are marked in the model schematic (Figure 1a). Uniformly sized through-holes are periodically distributed across the suspended LiNbO_3_ thin film; their diameter is D_1_. The IDT electrode is perpendicular to the Y direction of LiNbO_3_; the electrode width is W_e_, the electrode aperture is Le, the electrode pitch is G, the electrode duty ratio is C (its value is equal to W_e_/(W_e_ + G)), and the number of electrodes is N. The specific values of these key parameters are given in Table 1.

In a 2D Lamb wave resonator model (as Figure 1b), the resonant frequency can be given by Equation (1), where m and n are the mode orders in the vertical (*z*-axis) and longitudinal (*y*-axis) directions, respectively. V_t_ is the acoustic velocity in the direction of the LiNbO_3_ thin film thickness, V_L_ is the acoustic velocity in the direction perpendicular to the IDT electrode within the thin film plane, t is the LiNbO_3_ thin film thickness, and W_u_ is equal to the sum of the electrode pitch (G) and width (W_e_). Both V_t_ and V_L_ are closely related to the piezoelectricity of the LiNbO_3_. For the A1 mode resonators discussed in this paper, that is, m = 1, n = 1.
(1)$$f_{0}^{\mathrm{mn}}=\sqrt{{\left(m\frac{V_{t}}{2t}\right)}^{2}+{\left(n\frac{V_{L}}{2\mathrm{Wu}}\right)}^{2}}$$

## 3. Suppression of Spurious Modes by Through-Holes Structure

To validate the spurious modes-suppression by the through-holes, we calculated the admittance spectra of the A1 resonators with and without the through-holes by 3D multi-physics simulation, as shown in Figure 2. In these A1 resonators, W_e_ = 2 μm, G = 8 μm, and Le = 90 μm. Figure 2a–f show the admittance spectra for the through-holes diameter (D_1_) at 0 μm (i.e., no through-holes), 0.25 μm, 0.50 μm, 0.75 μm, 1.00 μm, and 1.25 μm, respectively.

In the absence of the through-holes (Figure 2a), the A1 resonator has explicit spurious modes in its admittance spectrum. Analysis of the field distribution of an obvious spurious mode (see inset) shows that the spurious mode presents a lateral third-order form of the A1 mode. Generally, it is called the lateral third-order A1 mode (i.e., m = 1, n = 3) or A1–3 mode, which may be produced by acoustic reflection at the edges of the IDT electrodes [43]. When the through-holes are relatively small, e.g., D_1_ = 0.25 μm (Figure 2b), the spurious mode is suppressed to a great extent but still presents. As the through-holes become larger, e.g., D_1_ = 0.50 μm (Figure 2c), the spurious mode is completely suppressed. When their diameter is continuously increased, e.g., D1 = 0.75 μm (Figure 2d) and D_1_ = 1.00 μm (Figure 2e), the spurious mode is all in suppression. With the further enlargement of the through-holes, e.g., D_1_ = 1.25 μm (Figure 2f), the spurious mode reappears but is still faint. In all the above cases, the A1 resonators’ admittance and the frequencies of the primary mode remain almost the same.

We further studied the spurious modes-suppression on the different spacing of the through-holes, as shown in Figure 3. Take the resonator in Figure 2e; for instance, Figure 3a–f show admittance spectra that the spacing between two adjacent through-holes is 7 μm, 8 μm, 9 μm, 10 μm, 11 μm, and 12 μm, respectively. It can be seen that when the spacing is small, e.g., 7 μm (Figure 3a), the spurious modes are already suppressed quite well, but there is still a little bit. They are completely suppressed with increased spacing, e.g., 8 μm to 10 μm (Figure 3b–d). When the spacing continues to increase, e.g., 11 μm and 12 μm (Figure 3e,f), the spurious mode reappears, but still only a little bit, much better than the situation without through-holes. Also, in these above cases, the A1 resonators’ admittance and the frequencies of the primary mode remain almost the same.

We also studied the influence of the position of the through-holes between the IDT electrodes on the spurious modes-suppression, as shown in Figure 4. Take the same resonator in Figure 2e as an instance, Figure 4a–f show the admittance spectra for the position of the through-holes moved upward by 0.5 μm, 1 μm, 1.5 μm and downward by 0.5 μm, 1 μm, and 1.5 μm, respectively. The results show that as long as the position of the through-holes is among the IDT electrodes, all unwanted spurious modes are completely suppressed.

In summary, for different hole sizes, spacings, and positions, the through-holes have a wide design window for the spurious modes-suppression of the A1 resonators. Also, compared with the LiNbO_3_ thickness (i.e., 300 nm in this study), the geometric depth-to-width ratio of the through-holes is quite small, making them convenient to fabricate.

## 4. Wide Applicability of the Through-Holes for IDT Designs

To implement well-functioning A1 filters, the IDT electrodes of the A1 resonators typically need to have various apertures and electrode duty ratios. Our through-holes for spurious mode-free resonators are well-applicable to different IDT designs. To demonstrate this, we calculated the admittance spectra of A1 resonators with identical through-holes for various IDT designs, as shown in Figure 5. Figure 5a–c show the admittance spectra for different electrode duty ratio (C). Figure 5d–f show the spectra for different IDT apertures. Specific values of the resonators’ parameters are labeled in the figures.

Without the through-holes, a spurious mode (A1–3 mode) at a higher frequency near the anti-resonant peak is present in all the resonators. In some resonators, another distinct spurious mode appears at a lower frequency near the main resonant peak. When the through-holes (D_1_ = 1.0 μm) are introduced, both spurious modes near the anti-resonance and main resonance peaks are well suppressed.

## 5. Wide Applicability of the Through-Holes for LiNbO_3_ Thickness

As mentioned in the introduction part, another advantage of the A1 resonators is that their operating frequencies are largely dependent on the thickness of the LiNbO_3_ thin film. By thinning the LiNbO_3_ thin film regionally, a single LiNbO_3_ thin film can have different thicknesses in different regions, thereby supporting A1 resonators with two or even multiple resonating frequencies. For variable thicknesses of the LiNbO_3_ thin films, the through-holes also have great applicability. Two randomly selected A1 resonators incorporating the through-holes were used as examples, and their admittance spectra were calculated for the variations of LiNbO_3_ thickness. The results are shown in Figure 6. Specific parameters of the resonators are marked in the figure. As can be seen, when thinning the LiNbO_3_ from 300 nm to 250 nm, no spurious mode appears in both two A1 resonators. At the same time, their resonant frequencies increase linearly.

For other A1 resonators with different IDT designs (e.g., in Figure 2 and Figure 5), we also calculated their admittance spectra after thinning the LiNbO_3_. The through-holes are all well capable of suppressing spurious modes.

## 6. Advantages of the Through-Holes for Device Fabrication and Stability

The through-holes also have important advantages for the processing and stability of the resonators (and subsequent filters), mainly involving two aspects.

Firstly, the preparation convenience of the through-holes itself, i.e., the introduction of the through-holes, does not add additional process steps. To prepare such Lamb wave resonators, it is usually necessary to etch several release windows on the piezoelectric (e.g., LiNbO_3_) thin film [31,42]. All through-holes can be fabricated simultaneously.

Secondly, and more importantly, the last step in the preparation of such resonators is to release the LiNbO_3_ thin film, i.e., to etch the box layer (e.g., SiO_2_) or substrate (Si) at the bottom of the piezoelectric thin film. The through-holes can significantly reduce the release process time and greatly reduce the invalid suspension area of the LiNbO_3_ thin film. Figure 7 shows the geometries of two resonators after the release (etching) process with and without the through-holes. Without through-holes, corrosive liquid or gas used to etch the SiO_2_/Si can only enter from peripheral release windows. Since the etching process is isotropic, while the LiNbO_3_ thin film with IDT is completely suspended, it also leads to a large invalid LiNbO_3_ suspension area around the resonator. This is detrimental to the mechanical and temperature stability of all Lamb wave resonators. When the through-holes are introduced, in addition to the original release windows, all the through-holes also act as release windows to enter corrosive liquids or gases. In this way, the LiNbO_3_ thin film with IDT can be suspended completely by enduring only a small amount of etching, which significantly reduces the area of the invalid LiNbO_3_ suspension area around the resonator, and greatly improves the stability of the resonator (and subsequent filters).

## 7. Conclusions

In this paper, we propose a general method for ideal LiNbO_3_ A1 resonators. By introducing subwavelength through-holes (with broad size requirement) among the IDT electrodes on the surface of the LiNbO_3_, all unwanted spurious modes can be completely suppressed. It is well-applicable under different IDT designs and LiNbO_3_ thicknesses.

For the fabrication of the A1 resonators and filters, no additional steps are required for the through-holes. Meanwhile, due to the existence of the through-holes, the etch (release) time for the substrate/box-layer (Si or SiO_2_) is drastically reduced. While improving the design flexibility, the invalid suspension area of the LiNbO_3_ thin film is also reduced, thus improving the compactness, mechanical stability, temperature stability, and power tolerance of the A1 resonators/filters. Our paper takes Z-cut LiNbO_3_ A1 resonators as an example. For other crystallographic orientations (e.g., 128° Y-cut), this design is also applicable. Also, for other Lamb waves resonators using different modes (e.g., other A, S, or SH modes) or different materials (e.g., AlN or AlScN), the through-holes may also have similar advantages, at least in terms of design convenience, processing convenience, and stability. Due to these advantages, we believe it will serve as a general solution and facilitate the practical application of Lamb wave resonators and filters.

## Figures and Tables

**Figure 1 micromachines-14-01341-f001:**
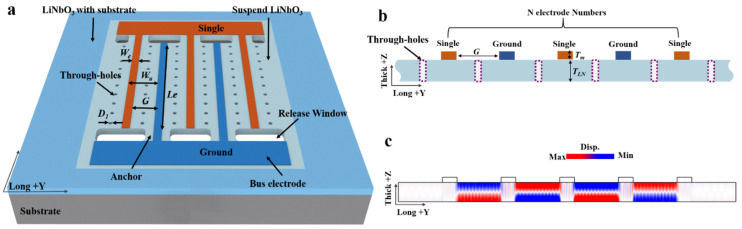
A1 resonators based on LiNbO_3_ film (including through-holes structure). (**a**) Global schematic diagram of the resonator. (**b**) Cross-sectional view of the resonator. (**c**) Out-of-plane displacement field distribution in the resonator at its resonant frequency.

**Figure 2 micromachines-14-01341-f002:**
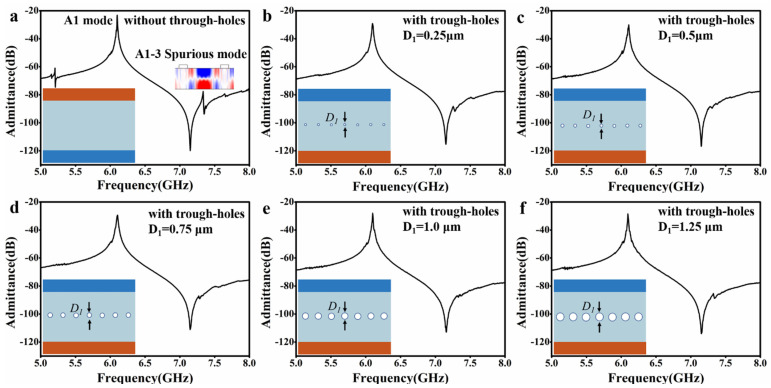
Through-holes diameter on spurious mode suppression. (**a**−**f**) The diameters of through-holes are 0 μm (i.e., no through-holes), 0.25 μm, 0.50 μm, 0.75 μm, 1.00 μm, and 1.25 μm, respectively.

**Figure 3 micromachines-14-01341-f003:**
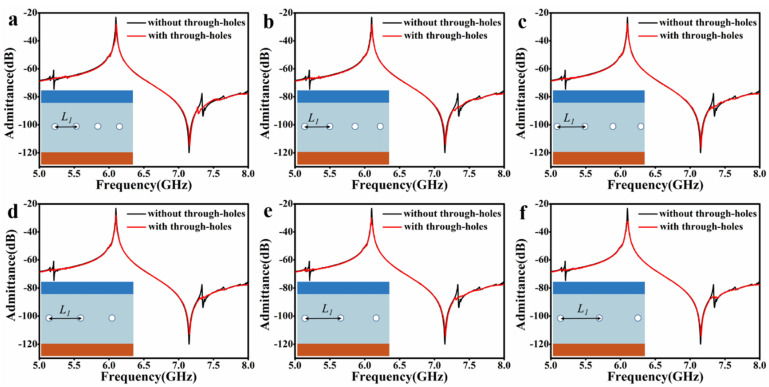
Through-holes’ spacings on spurious modes-suppression. (**a**–**f**) The spacings between two adjacent through-holes are 7 μm, 8 μm, 9 μm, 10 μm, 11 μm, and 12 μm, respectively.

**Figure 4 micromachines-14-01341-f004:**
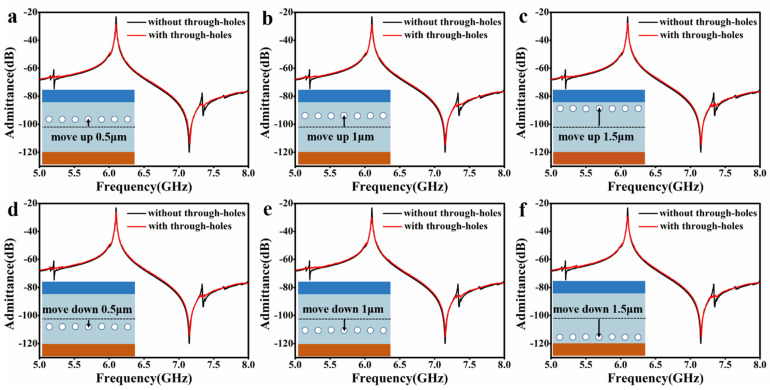
Through-holes position on spurious modes-suppression. (**a**–**c**) All the through-holes are moved upward 0.5 μm, 1 μm, and 1.5 μm, respectively. (**d**–**f**) All the through-holes are moved downward 0.5 μm, 1 μm, and 1.5 μm, respectively.

**Figure 5 micromachines-14-01341-f005:**
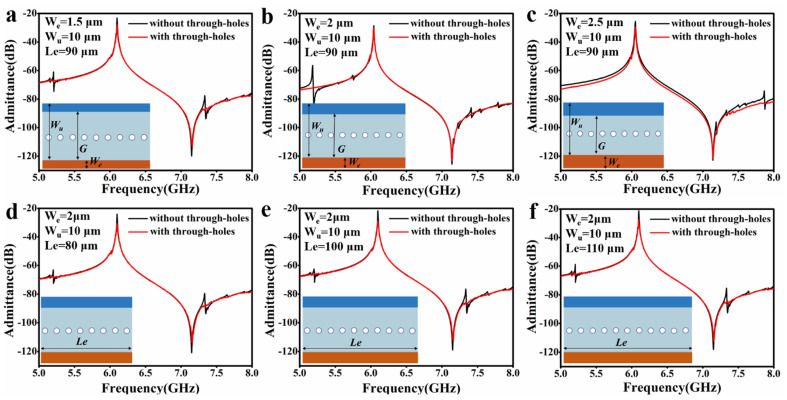
Spurious modes-suppression by the through-holes (D_1_ = 1 μm) with different IDT designs. (**a**–**c**): W_u_ = 10 μm, Le = 90 μm, W_e_ = 1.5 μm, 2.0 μm, and 2.5 μm, respectively. (**d**–**f**): W_u_ = 10 μm, W_e_ = 2.0 μm, Le = 80 μm, 100 μm, and 110 μm, respectively.

**Figure 6 micromachines-14-01341-f006:**
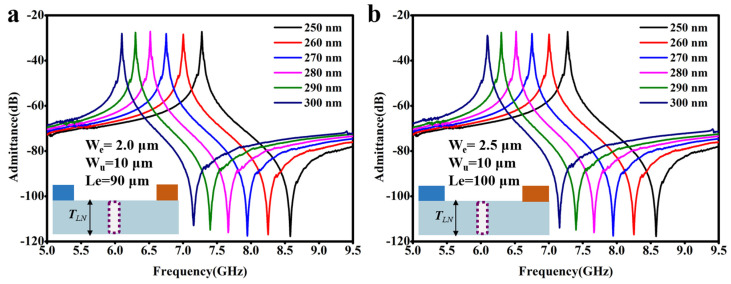
A1 resonators containing through-holes during thinning of the LiNbO_3_ thin film from 300 nm to 250 nm. IDT parameters of the two resonators are (**a**) W_u_ = 10 μm, W_e_ = 2 μm, Le = 90 μm and (**b**) W_u_ = 10 μm, W_e_ = 2.5 μm, Le = 100 μm.

**Figure 7 micromachines-14-01341-f007:**
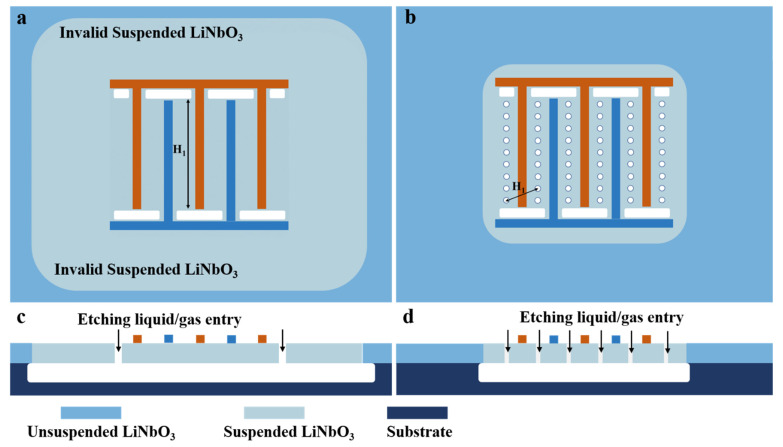
Geometries of the resonators after the release (etching) process without (**a**) top view (**c**) (cross-section view) and with (**b**) top view (**d**) (cross-section view) of the through-holes. H_1_ indicates the maximum etching/corrosion spacing.

**Table 1 micromachines-14-01341-t001:** Geometric parameter range of the A1 resonators studied in this paper.

Parameter	Value (μm)	Parameter	Value (μm)
Electrode width (W_e_)	1.5–2.5	LiNbO_3_ orientation	Z-cut
Electrode pitch (G)	7.5–8.5	Unit cell width (W_u_)	10
Electrode numbers (N)	5	LiNbO_3_ thickness (T_LN_)	0.25–0.30
Aperture (Le)	80–110	Au thickness (T_M_)	0.05
Electrode duty ratio (C)	0.15–0.25	Hole diameter (D_1_)	0.25–1.25

## Data Availability

Not applicable.

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
