# Peer review of "Through-Holes Design for Ideal LiNbO3 A1 Resonators"

_micromachines, 2023, doi:10.3390/mi14071341_

Round 1
Reviewer 1 Report
This article discusses the suppression of spurious modes in LiNbO3 A1 resonators via through-hole design, mainly using finite element analysis methods. The results are very enlightening and have verified their excellent suppression effect on spurious modes at the design level. I have the following questions and hope that the authors can provide a response.
1. In page 3, line 126 to 136, the authors wrote the following:
“At narrow electrode pitch, higher-order spurious modes almost certainly appear. In the case of large electrode pitch (e.g., above 10μm), the spurious mode can be partially suppressed.”
“Therefore, it is challenging to achieve spurious-mode-free A1 resonators with the larger electrode width and smaller electrode pitch.”
Does the size of the Electrode pitch need to be large or small in order to suppress spurious modes? I think these expressions are contradictory. Authors should reconsider it and make revise on it.
2. Equation (1) mentions the fundamental resonant frequency calculation for A1 mode, which does not consider the piezoelectric effect. Authors should make it clear to readers that the piezoelectricity is considered or not in the finite element calculation and whether it has an impact on the analysis results or not.
3. The author analyzed the impact of different hole sizes on spurious modes, but did not discuss the effects of hole position, spacing, etc. What are the considerations for setting these parameters in design. Can you provide a more optimal design for the hole placement?
4. What is the internal driving mechanism behind the suppression of spurious modes by holes. The article lacks an explanation of the physical mechanisms involved.
5. In the final conclusions, the author emphasized the convenience of the preparation process, but did not delve deeper into the number and position of holes to obtain more optimized results and determine whether it has an impact on the process.
6. If the process is not complex, the author should verify the correctness of the simulation results through experimental preparation.
Reviewer 2 Report
Authors propose a method to realize ideal lithium niobate (LiNbO3) A1 resonators by means of through holes. Thus they can suppress the unwanted spurious modes. Actually, authors vary the inductance of the structure by means of the through holes, so that the resonant frequency can be controlled. It could be better to add some explanations about the relation between the inductance and resonant frequency. Moreover, authors should emphasize the novelty or advantages of their method. In my humble opinion, the method is known but the authors use that approach for ideal LiNbO3 A1 resonators.
Quality of English is good. However, it would be better to recheck all the text one more time.
Reviewer 3 Report
1. The work done is not novel in terms of concept. Any through hole at the frequency of study should give a resonator characteristics.
2. The design provided is as similar to a substrate integrated waveguide.
3. The second harmonic will come at 2fo, (see fig.5) the graph provided doesn't show the suppression of the same. So how you prove that spurious modes are suppressed?
4. S21 shows pass band in 6- 7.5 GHz, but it is lossy ~ 20 dB. So how transmission is happening?
Reviewer 4 Report
look attached document

Round 2
Reviewer 1 Report
The manuscript can be accepted in present form.
Author Response
Our Response: We would like to thank the reviewer for his/her support of our work again.
Reviewer 3 Report
1. When ever a resonator is designed it serves the purpose of band pass or stop filter. Conventionally a short circuit or through hole should provide a band stop response in the spectrum thus suppressing the frequencies that it is designed for. So the through hole design should as per theory (many works done in the whole of EM spectrum) will suppress the spurious modes. Why the authors are unable to explain this theoretically? How is this novel?
2. If the design is similar to a substrate integrated waveguide, what is novel in the work. Authors failed to explain the similarity & differences based on the design.
3. In Fig. R 3-1 (a), the suppressed radiation is between 7 & 7.5 GHz. Mention if the resonance is at 6 GHz where will be the first, second & third order spurious modes as per calculation, is it suppressed as per the design & output.
Round 3
Reviewer 3 Report
Lot of improvement made in the paper. Do read through and do the minor corrections.